# Systematic Review and Meta-Analysis of Pleurectomy/Decortication versus Extrapleural Pneumonectomy in the Treatment of Malignant Pleural Mesothelioma

**DOI:** 10.3390/jcm11195544

**Published:** 2022-09-22

**Authors:** Federica Danuzzo, Sebastiano Maiorca, Gianluca Bonitta, Mario Nosotti

**Affiliations:** 1Department of Patho-Physiology and Transplantation, University of Milan, 20122 Milan, Italy; 2Foundation IRCCS Ca’ Granda Ospedale Maggiore Policlinico, 20122 Milan, Italy

**Keywords:** malignant pleural mesothelioma, mpm, mesothelioma, pleurectomy, pleurectomy/decortication (P/D), pneumonectomy, EPP

## Abstract

Objective: Malignant pleural mesothelioma (MPM) is a rare and aggressive malignant cancer for which there are poor treatment options. Extrapleural pneumonectomy (EPP) and pleurectomy decortication (P/D) are the two most used surgical procedures in patients with resectable disease. We reviewed the available literature in order to compare the overall survival and postoperative complications of EPP and P/D and to provide evidence for the best procedure in the treatment of MPM. Methods: We performed a systematic review of the literature, including studies from August 2018 to May 2022. The primary outcome was 5-year overall survival (OS) and the secondary outcomes were 30-day mortality, 90-day mortality and peri-operative complications. Results: Thirteen studies were considered, including a total of 1624 patients treated with EPP and 2147 treated with P/D. The estimated pooled HR showed a significant lower hazard for P/D compared to EPP in terms of OS (HR = 0.76; 95% CI from 0.62 to 0.94; *p* < 0.001). In 12 studies, the risk for 30-day mortality was lower for patients treated with P/D (RR = 0.49; 95% CI from 0.31 to 0.76; *p* = <0.01), whereas only five studies reported 90-day mortality, and no statistically significant difference between EPP and P/D was found (RR = 0.71; 95% CI from 0.47 to 1.07; *p* = 0.10). The OS restricted mean survival time difference meta-analysis (RMSDT) confirms the superior survival of P/D on the EPP, a superiority that increases from 0.54 months at one year to 4.23 at five years. The incidence of postoperative empyema, atrial fibrillation, bleeding and bronchopleural fistula was significantly increased in the EPP group except for prolonged air leakage, which is only characteristic of P/D. Conclusions: Using two different statistical methods, this meta-analysis suggests that long-term survival after surgical treatment for resectable MPM is greater for patients undergoing P/D. Long-term survival had never been previously analyzed with appropriate tests; on the contrary, our result is consistent with the previous meta-analyses and reinforces the evidence of lower 30-day mortality and the prevalence of postoperative complications in P/D versus EPP patients. The recent introduction of innovative therapeutic schemes, both adjuvant and neoadjuvant therapy, keeps the discussion on surgical strategy open and will require new studies.

## 1. Introduction

Malignant pleural mesothelioma (MPM) is a rare malignant cancer for which there are poor treatment options. MPM was widely considered as a medical curiosity until 1960, when Wagner reported 33 cases of MPM in South African asbestos mine workers [1]. Subsequent studies have confirmed that exposure to asbestos was the main risk factor for this insidious tumor. MPM is a disease characterized by a large latency period between asbestos exposure and clinical manifestation; this is why, despite the asbestos ban, the incidence of the disease in the world population is increasing. In cases where disease is unresectable, palliative chemotherapy with platinum-based chemotherapy associated with pemetrexed or palliative radiotherapy are typically used. In those patients where the disease is resectable, the two most commonly used surgical procedures are extrapleural pneumonectomy (EPP) and pleurectomy decortication (P/D). P/D is defined as the removal of the parietal and visceral pleura, ranging from minimally-invasive partial pleural resection to radical pleural excision involving the pericardium and diaphragm. The Mesothelioma Interest Group and the International Association for the Study of Lung Cancer, published a Consensus Report that classified the pleurectomy-decortication into three defined surgical procedures: partial pleurectomy (partial removal of parietal and/or visceral pleura), pleurectomy-decortication (parietal and visceral pleurectomy without resection of the diaphragm or pericardium), and extended pleurectomy-decortication (pleurectomy-decortication plus the resection of the pericardium and hemidiaphragm). EPP is defined as the en bloc removal of the lung affected by disease, visceral and parietal pleura, pericardium and hemidiaphragm.

EPP, as well as the various types of P/D, can be combined with a large range of adjuvant therapies (chemotherapy, radiotherapy, intrapleural chemotherapy, photodynamic therapy), increasing the dispersion of results in this rare disease. Some meta-analyses have been carried out to clarify which of the different therapeutic strategies may be the most effective [2,3,4]. Although the methodology adopted in these meta-analyses has been criticized, there was a unanimous indication that P/D is the suggested surgery for potentially resectable pleural mesothelioma [5]. 

Considering the constant publication of new trials, it is common opinion that the search for new evidence to adjourn previous systematic review is worthwhile [6]. Indeed, the purpose of this study is to analyze and update existing data comparing the overall survival and postoperative complications of EPP and P/D to provide evidence for the best procedure in the treatment of MPM.

## 2. Materials and Methods

### 2.1. Search Strategy and Articles Selection

We performed a systematic review of the current literature in accordance with Reporting Items for Systematic Reviews and Meta-Analyses checklist guidelines (PRISMA) [7] and without any ethical approval requirement (Appendix A). As a starting point, we considered the articles included in the 2018 meta-analysis by Magouliotis et al. [4], then, we used PubMed (Medline) and Cochrane Central Library (last date of search: 6 May 2022) with a combination of the following MeSH (Medical Subject Headings) terms: (“Surgical Procedures, Operative” [MESH]) AND “Mesothelioma/surgery” [Majr]. Finally, we performed a manual search with the following MeSH terms: “malignant pleural mesothelioma”, “mpm”, “mesothelioma”, “pleurectomy”, “pleurectomy/decortication”, “P/D”, “pneumonectomy” and “EPP”. 

The inclusion criteria were original reports conducted on human subjects and written in the English language; at least 10 patients for either P/D or EPP groups; studies reporting long-term survival data or Kaplan–Meier survival curves. 

The exclusion criteria were articles not written in English; surgical technique in contrast with the consensus report of the IASLC-IS Committee [8]; studies not reporting the a priori defined primary outcome; limited sample size.

Duplicate articles were excluded, and the reference lists of each included article were reviewed for additional studies or to identify citations. 

Two reviewers (FD, SM) independently screened the titles and abstracts of the studies of the bibliographic search. Disagreements were resolved through discussion and, if required, the consultation of a third review author (MN).

### 2.2. Data Extraction

For each study included in our meta-analysis, the following data were extracted and collected: authors, year of publication, country, study design, type of P/D approach according to IASLC-ISC criteria (extended or not P/D), stage of disease, neoadjuvant or adjuvant treatment, number of patients, sex, mean age, survival, postoperative outcomes and complications. 

All data were collected independently by two investigators (F.D., S.M.) and compared at the end of the reviewing process. A third author (M.N.) reviewed the database to identify discrepancies.

### 2.3. Outcomes of Interest

The primary outcome was 5-year overall survival (OS). The secondary outcomes were 30-day mortality, 90-day mortality and peri-operative complications such as atrial fibrillation, empyema, hemorrhage, pulmonary embolism, myocardial infarction and prolonged air leaks.

### 2.4. Statistical Analysis

The results of the systematic review were summarized qualitatively into a frequentist meta-analysis of pooled hazard ratio (HR) and risk ratio (RR). The inverse-variance random-effects meta-analysis was performed by conventional methods using DerSimonian–Laird estimator for estimate the between-study variance (τ2) [9]. The restricted maximum-likelihood and Q-profile method were performed in order to estimate τ2. Statistical heterogeneity was evaluated by the Cochran Q-test and I2 index: a value of 25% or smaller was defined as low heterogeneity, between 50 and 75% as moderate heterogeneity, and 75% or larger as high heterogeneity [10].

Wald-type 95% confidence intervals (95% CI) were computed for the pooled measures. The prediction interval for the treatment effect of a new study was calculated according to Borenstein [11]. As the sample size is not the same in all studies, we gradually removed a small sample size to perform a sensitivity analysis to assess the stability of the results (one-leave-out test). Small studies and publication bias effects were assessed by a funnel plot visual inspection and Egger tests for outcomes reported in more than 9 studies.

The individual patient time-to-event data were reconstructed from Kaplan–Maier curves according to Guyot [12]. The Kaplan–Meier curves were digitalized using the Get Data Graph Digitizer software (http://getdata-graphdigitizer.com, accessed on 6 May 2022). The HR and relative standard errors were computed by the univariable Cox regression model, and the proportional hazard assumption was checked by means of the diagnostics based on the scaled Schoenfeld residuals [5]. A meta-analysis of restricted mean survival time difference (RMSTD) was performed using a random effect multivariate meta-analysis borrowing strength across time-points with a within-trial covariance matrix derived by a bootstrap method with 1000 iterations [13]. Since the sample sizes varied among the studies, we performed the one-leave-out sensitivity test to verify the robustness of the results [14]. A Z-score test was performed. Two-sided p-values were considered statistically significant when less than 0.05 and the CIs were computed at 95%. We used EPP as reference in all statistical analyses. All statistics and graphs were carried out using the R software application (version 3.2.2; R Foundation, Vienna, Austria).

### 2.5. Quality and Publication Bias Assessment

The Newcastle–Ottawa Quality Assessment Scale (NOS) [15] was used for assessing the quality of non-randomized studies (Appendix A). This is a ‘star system scale’ ranging from zero to nine stars. Each study is judged on eight items categorized in three domains: the selection of the study groups, the comparability of the groups and the ascertainment of either the exposure or outcome of interest for case-control or cohort studies, respectively. The studies included in this meta-analysis were awarded a score equal to or higher than five.

## 3. Results

### 3.1. Articles Selection and Patient Demographics

The flow diagram of the literature search, according to the PRISMA checklist, is shown in Figure 1.

Among the 111 articles retrieved in PubMed, 13 studies were included in the quantitative and qualitative synthesis [16,17,18,19,20,21,22,23,24,25,26,27,28]. The study design was prospective in one study [22] and retrospective in 12 [16,17,18,19,20,21,23,24,25,26,27,28]; two retrospective studies applied the propensity score match analysis [21,28]. The included studies were published between 2003 and 2022 and were conducted in the UK [22,23,26], Italy [16,25], South Africa [17], the USA [18,27,28], Switzerland [21], Japan [19,24] and Singapore [20]. Ten studies were cohort retrospective studies [16,17,18,19,20,23,24,25,26,27]; of these, eight were monocentric studies [17,18,19,20,23,24,25,26], one multicentric [16] and one based on a national database [27]. Among the ten retrospective studies, four covered a time interval of 20 years or more [16,17,23,24]. Two studies were retrospective cohort studies with propensity score matching and both were monocentric and covered a time interval between 16 and 19 years [21,28]. One study was a prospective nonrandomized interventional study, which was performed at a single center. The time interval was only seven years but the decision on which surgery to perform was taken on clinical bases [22]. Similarly, clinical criteria were used for the selection of the surgical procedure in all other selected studies. The total number of patients was 3771; 2147 were treated with either P/D or extended P/D and 1624 were treated with EPP. The main characteristics of the studies included in our meta-analysis are summarized in the Table 1.

### 3.2. Meta-Analysis: Primary Outcomes

The estimated pooled HR showed a significantly lower hazard for P/D compared to EPP in terms of OS (HR = 0.76; 95% CI from 0.62 to 0.94; *p* < 0.001), considering the random effects model on 13 studies including 3771 patients. Heterogeneity was moderate (I2 = 74.3%, CI from 56.6 to 85.1%, *p* < 0.001). The forest plot for OS is shown in Figure 2. A sensitivity analysis showed the robustness of results in terms of point estimation, confidence intervals and heterogeneity. The visual inspection of Schoenfeld residuals and the related global Schoenfeld test for HR calculation showed evidence of proportional hazard assumption violation in two studies. The funnel plot (Figure 3) and Egger’s test showed that publication bias could not be excluded.

An evaluation of the restricted mean survival time difference based on Kaplan–Meier curves for OS was possible for all the selected studies. All the studies had a minimum follow-up of 3 years, twelve studies had a follow-up of 4 years, and nine studies had a follow-up of 5 years. The RMSTD analysis with the time horizons is presented as a graphic (Figure 4) and detailed in Table 2. There was a significant RMSTD for P/D versus EPP at 12, 24, 36, 48 and 60 months of 0.54 (95% CI from 0.03 to 1.05; *p* = 0.038); 1.69 (95% CI from 0.4 to 2.98; *p* = 0.010); 2.56 (95% CI from 0.63 to 4.49; *p* = 0.09); 3.36 (95% CI from 0.78 to 5.94; *p* = 0.01); and 4.23 months (95% CI from 1.0 to 7.46; *p* = 0.01), respectively.

### 3.3. Meta-Analysis: Secondary Outcomes

Short-term mortality. Data regarding the 30-day and 90-day perioperative mortality are reported in Table 3. Based on 12 studies, the risk for 30-day mortality was lower for patients treated with P/D: RR = 0.49; 95% CI from 0.31 to 0.76; *p* = <0.01 (Figure 5 and Figure 6). Only five studies reported 90-day mortality; no statistically significant difference between EPP and P/D was found: RR = 0.71; 95% CI from 0.47 to 1.07; *p* = 0.10 (Figure 7).

Perioperative complications. Ten trials reported the total postoperative complications rate; the pooled risk for complications was significantly lower in patients who underwent P/D (RR = 0.64; 95% CI from 0.45 to 0.90; *p* < 0.01) (Figure 8). Namely, the risk of perioperative morbidity was significantly lower after P/D compared to EPP in terms of post-operative empyema (RR = 0.12; 95% CI from 0.05 to 0.30; *p* < 0.01), atrial fibrillation (RR = 0.39; 95% CI from 0.20 to 0.76; *p* < 0.01) and hemorrhage (RR = 0.35; 95% CI from 0.17 to 0.71; *p* < 0.01) (Figure 9A–C).

Post-operative prolonged air leaks were present only in the P/D group; the pooled prevalence was 0.20 (CI from 0.07 to 0.47, I2 = 94%) (Figure 9D).

No significant differences regarding the post-operative myocardial infarction (RR 1.19; 95% CI from 0.25 to 5.72; *p* = 0.87) and pulmonary embolism (RR 0.51; 95% CI from 0.18 to 1.47; *p* = 0.27) were found between the two procedures (Figure 9E,F).

## 4. Discussion

The present study reviewed published data comparing pleurectomy-decortication with extrapleural pneumonectomy for patients with resectable MPM. All the available evidence in the current literature were updated to May 2022. Despite P/D procedures varying significantly in surgical technique, we presumed that all the trials comparing EPP with P/D incorporated surgery with therapeutic intent; therefore, we avoided only selecting trials including exclusively extended P/D. This decision made it possible to triple the number of included patients, a not inconsiderable aspect in a rare disease.

Focused on overall survival, our analysis seems to point out improved long-term survival in P/D patients. Five years after surgery, the estimated pooled HR of P/D vs EPP is 0.76 considering 3771 patients. The time-to-event curves analyzed by Cox proportional hazards regression was never previously calculated in meta-analyses of surgical treatment for MPM. In the 2018 meta-analysis published by Magouliotis and collaborators, the authors calculated the odds ratio at different time points after surgery without finding significant differences between the two types of treatment [4]. Other meta-analyses did not calculate long-term survival due to a lack of data [2,3]. It is well known that the hazard ratio is somewhat more indicative and appropriate than the odds ratio because it correspond to instantaneous risk over the entire study period and is less affected by selection bias. Indeed, our HR calculation for OS returned a value indicating moderate heterogeneity. Despite these positive considerations, we urge taking our result with caution because in the calculation of HR we noted a violation of the assumption of the proportionality of risk in two studies. For this reason, we added the OS restricted mean survival time difference meta-analysis. This alternative method of comparison confirms the superior survival of P/D on the EPP, a superiority that increases from 0.54 months at one year to 4.23 at five years. Like HR, RMSTD was never previously calculated for a meta-analysis of survival after MPM surgery. We decided to introduce this method because it does not require the assumption of proportionality, as does the Cox regression. In fact, it is likely that the early postoperative surgical course can have a significant impact on the proportionality of hazard, since the two surgical techniques are extremely different in terms of biological impact (pneumonectomy versus no-pneumonectomy). Moreover, RMSTD has a better clinical interpretation than HR. For example, our RMSTD meta-analysis returned a survival difference of three and a half months in favor of D/P after 4 years of surgery. This result is extremely understandable to both clinicians and patients.

The results of the present meta-analysis suggest that P/D was associated with significantly lower 30-day mortality rates compared to EPP. The risk of death at 30 days for a patient undergoing P/D is substantially halved compared to a patient undergoing EPP. This result is reinforced by the good number of studies included, as well as by the heterogeneity test, which returned a value of zero. Our result is consistent with the previous meta-analyses; therefore, we can speculate that new trials will hardly be able to modify this point. In contrast, the meta-analysis of 90-day mortality did not reveal a significant difference between the two procedures. We must emphasize that this calculation includes only five studies and, consequently, the result will change with the publication of new trials.

Previous meta-analyses reported a significantly lower risk of postoperative complications in patients treated with P/D [2,3,4]. Our analysis confirms the lower morbidity of P/D, although there is high heterogeneity. If we exclude the prolonged air leakage that is only characteristic of P/D, this procedure is significantly less burdened by complications such as empyema, atrial fibrillation and bleeding. In particular, our meta-analysis of the pooled risk ratio for empyema and hemorrhage has low heterogeneity and a prediction interval definitively in favor of P/D.

The present study has several limitations. Most selected trials were retrospective observational studies; therefore, the choice between P/D and EPP was made on clinical considerations such as patients’ age, comorbidities, pathological characteristics and stage. Each of these parameters probably had an independent impact on survival, introducing selection biases. Moreover, it is highly probable that clinical follow-ups were dissimilar in different centers by introducing biases both on survival and on the categorization of complications. In addition, a publication bias was noted for the main result (Figure 3); this bias can be attributed to the previously mentioned heterogeneity, but also to the quality of the studies, the outcome measures and finally, the case. We mitigated these discrepancies by statistic methods as best we could. Furthermore, it should be noted that our meta-analysis includes two different surgical procedures, P/D and extended P/D, each of which may have been performed with different protocols by the different surgical groups. A striking example is the trial published by Lang-Lazdunski and collaborators; the authors state that of the 54 patients subjected to P/D, only seven had the resection of the pericardium or diaphragm, while five had typical lung resections [22]. Moreover, the lack of indication of the surgical timing as well as the lack of clarification of when the follow-up started certainly contributed to the marked heterogeneity. Only the large number of patients included and the use of appropriate statistical tests allowed us to mitigate these inequalities. Finally, it is certain that the patients included in this meta-analysis received highly diversified adjuvant and/or neo-adjuvant treatments, since during the studied periods the FDA approved the pemetrexed. The first trials of induction therapy appeared around 2010, and immune checkpoint inhibitors were recently introduced in the MPM treatment. In the absence of data specifically referring to the individual treatment of each patient, adjuvant and neo-adjuvant treatments were not subjected to analysis.

The surgical treatment of MPM remains one of the most discussed topics in thoracic surgery. Since the debated results of the MARS randomized trial, which questioned the effectiveness of EPP in the treatment of MPM, surgeons have had to choose which surgery to propose to patients without the support of robust literature [29]. This meta-analysis reinforces evidence of lower 30-day mortality and the lower prevalence of postoperative complications in P/D versus EPP patients. By examining long-term survival for the first time, our meta-analysis returns a limited but significant benefit for patients treated with P/D. Much work remains to establish the optimal treatment for MPM; well-designed trails will be essential to understand which surgical strategy can be integrated with innovative medical treatments such as immune checkpoint inhibitors or antiangiogenic drugs.

## Figures and Tables

**Figure 1 jcm-11-05544-f001:**
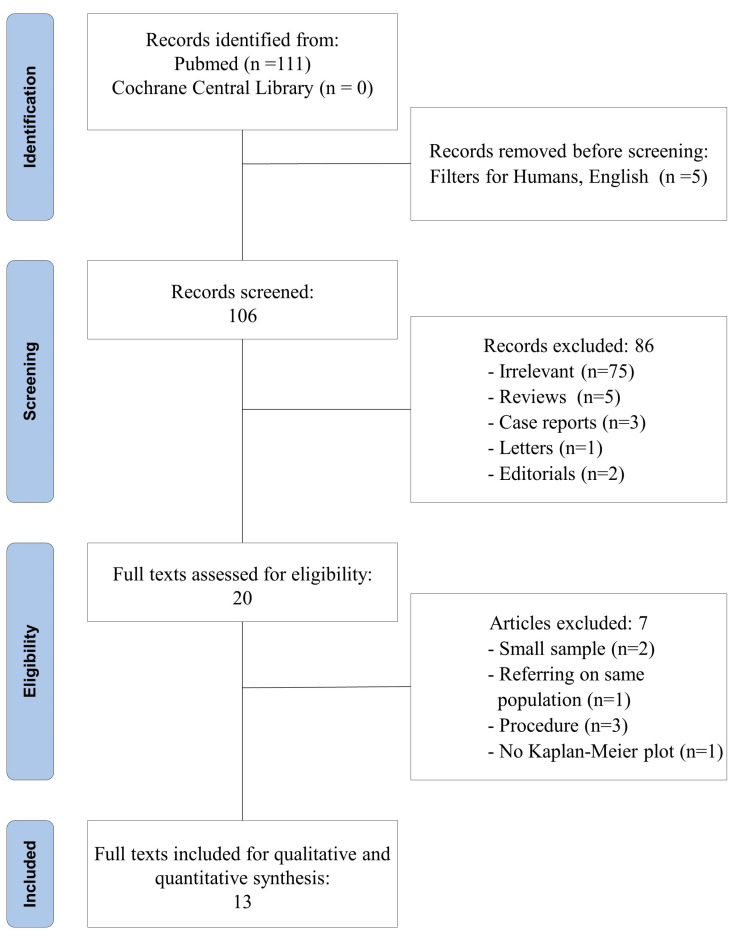
PRISMA flow diagram for study review and inclusion.

**Figure 2 jcm-11-05544-f002:**
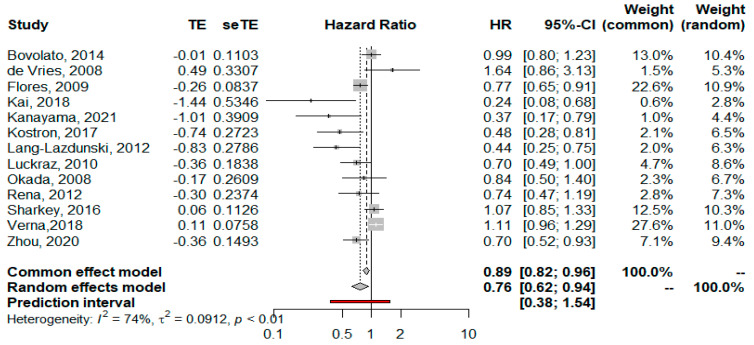
Forest plot of the hazard ratio (HR) for overall survival of all-cause mortality after pleurectomy/decortication (P/D) versus extrapleural pneumonectomy (EPP) in the treatment of malignant pleural mesothelioma [16,17,18,19,20,21,22,23,24,25,26,27,28]. The pooled HR with 95% confidence interval (CI) using a random effects model is 0.76, in favor of P/D. The test of heterogeneity between the studies within a subgroup is shown below in the summary statistics.

**Figure 3 jcm-11-05544-f003:**
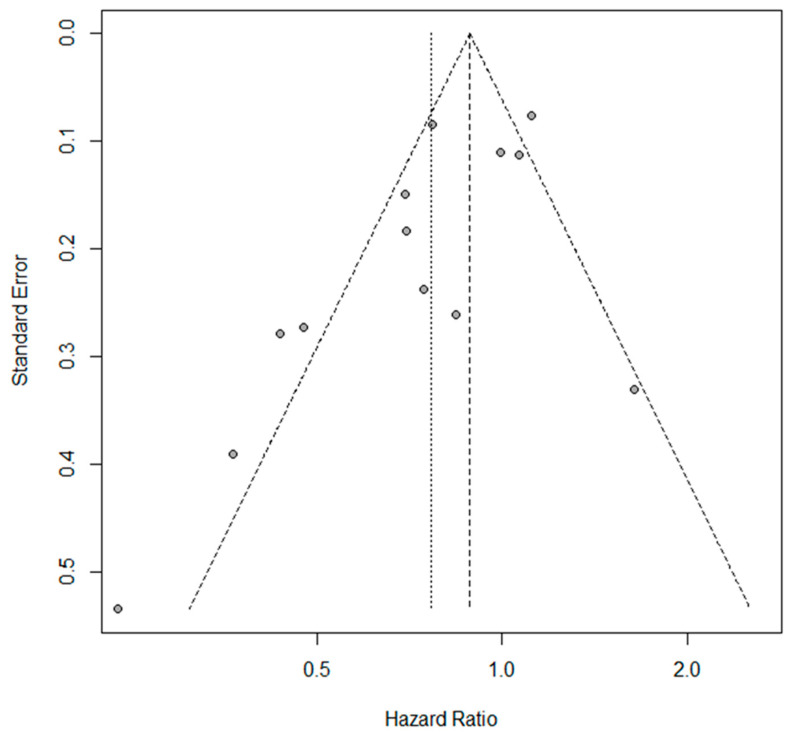
Funnel plot for overall survival. The dots asymmetric distribution indicates possible publication bias.

**Figure 4 jcm-11-05544-f004:**
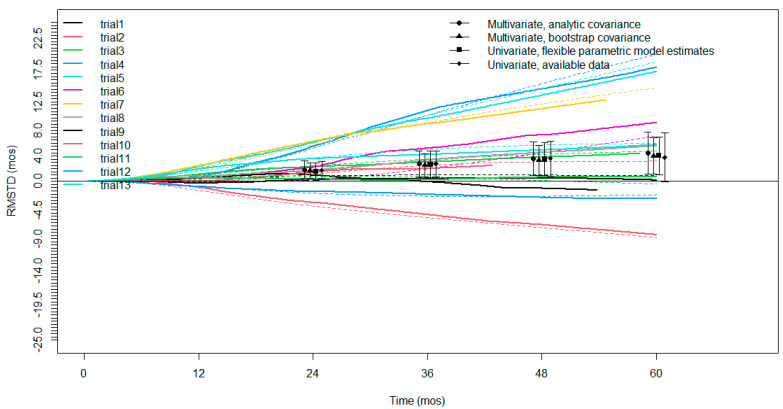
Restricted mean survival time difference (RMSTD) for all selected studies. Each color represents a single study. For every color there are dashed (PD) and continued (EPP) tracts. The pooled RMSTD with relative 95% confidence interval using different statistical estimations is represented by black points (triangle, circle, square and rhombus).

**Figure 5 jcm-11-05544-f005:**
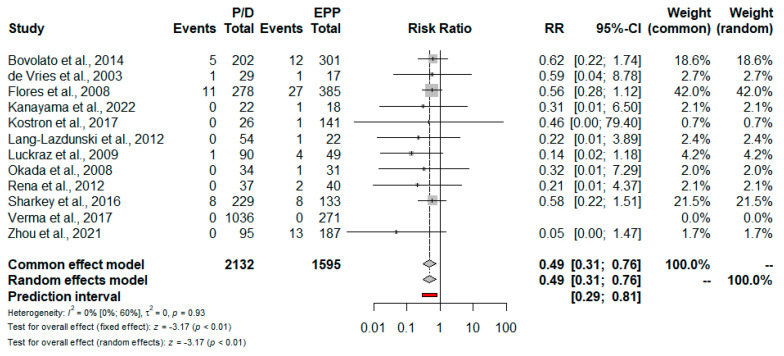
Forest plot of the relative risk (RR) for 30-day mortality after pleurectomy/decortication (P/D) versus extrapleural pneumonectomy (EPP) [16,17,18,20,21,22,23,24,25,26,27,28]. The pooled RR with 95% confidence interval (CI) using a random effects model is 0.49, in favor of P/D. The I^2^ index shows low heterogeneity.

**Figure 6 jcm-11-05544-f006:**
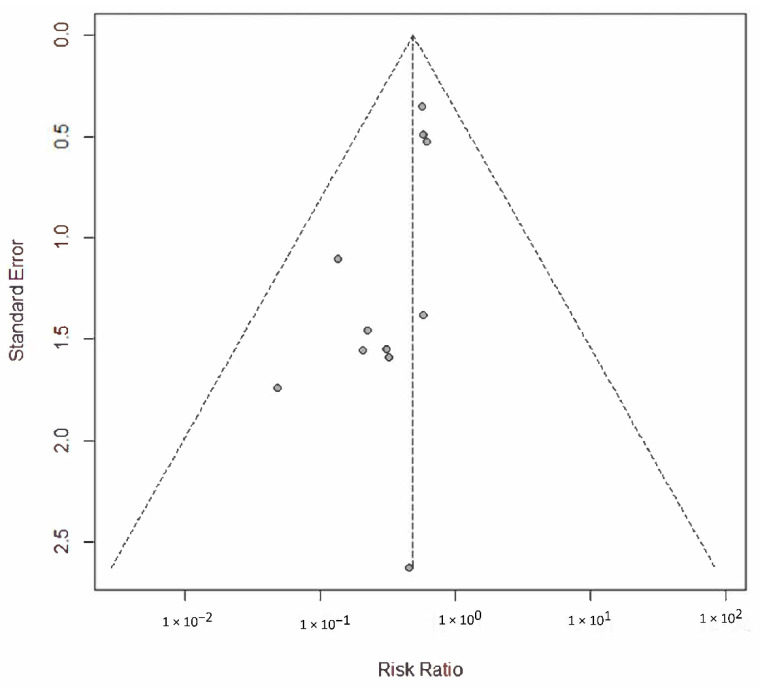
Funnel plot for 30-day mortality. The dots symmetric distribution indicates low publication bias.

**Figure 7 jcm-11-05544-f007:**
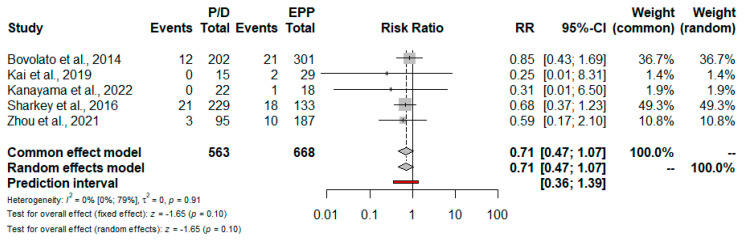
Forest plot of the relative risk (RR) for 90-day mortality after pleurectomy/decortication (P/D) versus extrapleural pneumonectomy (EPP) [16,19,20,26,28]. The pooled RR using a random effects model does not differ between the two treatments. The test of heterogeneity between the five studies within a subgroup is shown below in the summary statistics.

**Figure 8 jcm-11-05544-f008:**
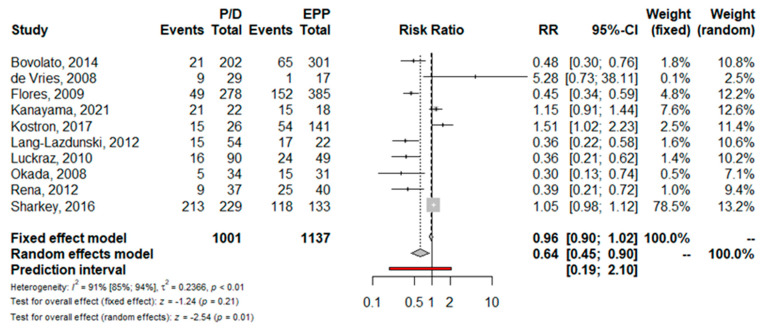
Forest plot of the relative risk (RR) for total postoperative complications after pleurectomy/decortication (P/D) versus extrapleural pneumonectomy (EPP) [16,17,18,20,21,22,23,24,25,26]. The pooled RR using a random effects model shows a significative lower risk of complication for P/D. The test of heterogeneity between the 10 studies within a subgroup returns a high value.

**Figure 9 jcm-11-05544-f009:**
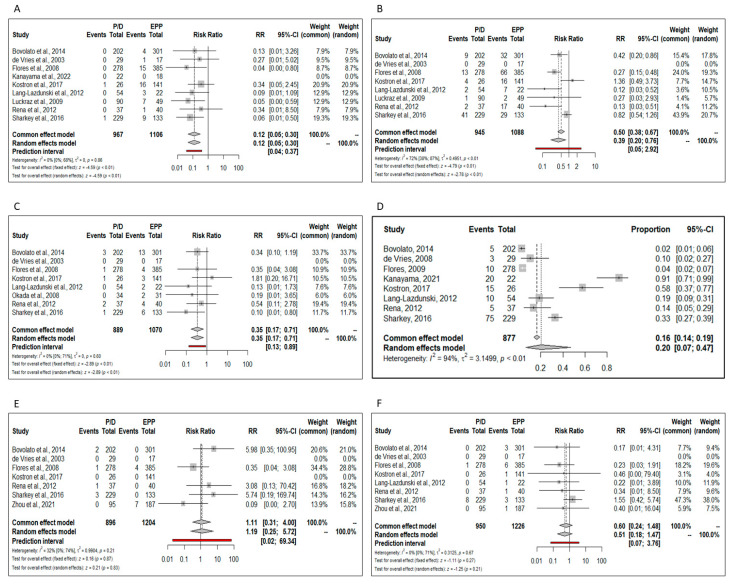
Forest plots of the relative risk (RR) for major perioperative complications. (**A**) Empyema [16,17,18,20,21,22,24,25,26,28]. (**B**) Atrial fibrillation. (**C**) Hemorrhage. (**D**) Prolonged air leakage. (**E**) Myocardial infarction. (**F**) Pulmonary embolism.

**Table 1 jcm-11-05544-t001:** Summary of study characteristics.

Authors, Year (Reference Number)	Country	Study Design	P/D Approach	Patients, *n*		Female, *n* (%)		Mean Age (Range)		Stage	NOS	Neoadjuvant/Adjuvant
				EPP	P/D	EPP	P/D	EPP	P/D			
Bovolato et al., 2014 [16]	Italy	R	P/D	301	202	76 (25.3)	53 (26.2)	58.7 (33–78)	62.5 (30–87)	I: 9.5%; II: 27.6%; ≥III: 19%; ND: 43.7% (ND)	6	adjuvant CT RT
de Vries et al., 2003 [17]	South Africa	R	P/D	17	29	–	–	–	–	I–III **	5	adjuvant CT RT
Flores et al., 2008 [18]	USA	R	Extended P/D	385	278	69 (18)	58 (21)	60	63	I: 2%; II: 95%; III: 24%; IV: 16%; ND: 48% ***	7	adjuvant CT RT
Kai et al., 2019 [19]	Japan	R	Extended P/D	29	15	0 (0)	0 (0)	64 (42–73)	66 (48–72)	I–IV *	6	neoadjuvant + adjuvant
Kanayama et al., 2022 [20]	Singapore	R	Extended P/D	18	22	1 (5.6)	3 (13.6)	64.0 (51–77)	69.5 (54–82)	IA–IIIB *	6	adjuvant
Kostron et al., 2017 [21]	Switzerland	R PSM	P/D and extended P/D	141	26	16 (11)	1 (4)	61 (36–73)	66 (34–77)	I–IV *	7	neoadjuvant CT
Lang-Lazdunski et al., 2012 [22]	UK	P	P/D and extended P/D	22	54	2 (8.1)	7 (13)	62 (52–68)	62.5 (45–74)	I–IV *	6	neoadjuvant + adjuvant
Luckraz et al., 2009 [23]	UK	R	P/D	49	90	0 (0)	4 (12)	57.8 (12.6)	63.5 (9.1)	I–III **	5	neoadjuvant + adjuvant
Okada et al., 2008 [24]	Japan	R	Extended P/D	31	34	4 (13)	3 (9)	60 (35–72)	60 (37–78)	I–IV *	6	neoadjuvant + adjuvant
Rena et al., 2012 [25]	Italy	R	P/D	40	37	29 (27)	6 (32)	56 ± 11 °	58.5 ± 9.5 °	I, II (ND)	6	neoadjuvant + adjuvant
Sharkey et al., 2016 [26]	UK, Italy	R	Extended P/D	133	229	19 (14)	33 (14)	57 (14–70)	65 (42–81)	I: 4%; II: 14%; III: 57%; IV: 25% *	6	neoadjuvant + adjuvant
Verma et al., 2017 [27]	USA	R	Extended P/D	271	1036	60 (22)	217 (21)	65 (58–71)	69 (62–76)	I–IV (ND)	7	adjuvant CT RT
Zhou et al., 2021 [28]	USA	R PSM	ND	187	95	32 (17.1)	24 (26.3)	61 (56–67)	65 (61–71)	I–III (ND)	6	adjuvant RT

EPP: extrapleural pneumonectomy; P/D: pleurectomy-decortication; NOS: Newcastle–Ottawa Quality Assessment Scale, number of stars; R: retrospective study; P: prospective study; PSM: propensity score matching; ND: non-defined; ° standard deviation; * IMIG 8th edition; ** Butchart and Brigham staging systems; *** AJCC Cancer Staging Handbook 6th edition.

**Table 2 jcm-11-05544-t002:** Restricted mean survival time difference at different time horizons for pleurectomy-decortication versus extrapleural pneumonectomy.

Time Horizon	No. Trials	RMSTD (Months)	SE	95% CI	*p* Value
12 months	13	0.5	0.3	0.03 to 1.05	0.038
24 months	13	1.7	0.6	0.4 to 2.98	0.010
36 months	13	2.6	1.0	0.63 to 4.49	0.009
48 months	12	3.4	1.3	0.78 to 5.94	0.01
60 months	9	4.2	1.6	1.0 to 7.46	0.01

RSMTD: restricted mean survival time difference; SE: standard error; CI: confidence interval. The pooled RMSTD values suggested a mild estimated advantage in the overall survival of pleurectomy-decortication on extrapleural pneumonectomy at the indicated time points.

**Table 3 jcm-11-05544-t003:** Operative mortality and postoperative complications rates.

Authors, Year (Reference Number)	Nunber of Patients		30 Days Mortality		90 Days Mortality		Total Complications		Atrial Fibrillation		Bleeding		Pulmonary Embolism		Miocardial Infarction		Empyema		Prolongued Air Leaks	
	EPP	P/D	EPP	P/D	EPP	P/D	EPP	P/D	EPP	P/D	EPP	P/D	EPP	P/D	EPP	P/D	EPP	P/D	EPP	P/D
Bovolato et al., 2014 [16]	301	202	4.1%	2.6%	6.9%	6%	21.6%	10.4%	10.6%	4.5%	4.3%	1.5%	1%	0%	0%	1%	1.3%	0%	0%	1.3%
de Vries et al., 2003 [17]	17	29	5.8%	3.8%	N.A.	N.A.	5.9%	31%	0%	0%	0%	0%	0%	0%	0%	0%	5.9%	0%	0%	1%
Flores et al., 2008 [18]	385	278	7%	4%	N.A.	N.A.	39.4%	17.6%	17.1%	4.7%	1%	0.4%	1.6%	0.4%	1%	0.4%	3.9%	0%	0%	3.6%
Kai et al., 2019 [19]	29	15	N.A.	N.A.	6.9%	0%	N.A.	N.A.	N.A.	N.A.	N.A.	N.A.	N.A.	N.A.	N.A.	N.A.	N.A.	N.A.	N.A.	N.A.
Kanayama et al., 2022 [20]	18	22	5.6%	0%	5.6%	0%	83.3%	95.5%	N.A.	N.A.	N.A.	N.A.	N.A.	N.A.	N.A.	N.A.	0%	0%	0%	91%
Kostron et al., 2017 [21]	141	26	2%	0%	8%	0%	38%	58%	11.3%	15.4%	2.1%	3.8%	0.7%	0%	0%	0%	11.3%	3.8%	0%	57.7%
Lang-Lazdunski et al., 2012 [22]	22	54	4.5%	0%	N.A.	N.A.	68%	27.7%	31.8%	3.7%	9%	0%	4.5%	0%	N.A.	N.A.	13.6%	0%	0%	45.5%
Luckraz et al., 2009 [23]	49	90	8.2%	1.1%	N.A.	N.A.	48.9%	17.7%	4%	1%	N.A.	N.A.	N.A.	N.A.	N.A.	N.A.	14.3%	0%	N.A.	N.A.
Okada et al., 2008 [24]	31	34	3.2%	0%	N.A.	N.A.	48%	15%	N.A.	N.A.	6.5%	0%	N.A.	N.A.	N.A.	N.A.	N.A.	N.A.	N.A.	N.A.
Rena et al., 2012 [25]	40	37	5%	0%	N.A.	N.A.	62%	24%	42.5%	5.4%	10%	5.4%	2.5%	0%	0%	2.5%	2.7%	0%	0%	13.5%
Sharkey et al., 2016 [26]	133	229	6%	3.5%	13.5%	9.2%	88.7%	93%	21.8%	17.9%	4.5%	0.4%	2.3%	3.5%	0%	1.3%	6.8%	4.4%	0%	32.8%
Verma et al., 2017 [27]	271	1036	5%	5%	N.A.	N.A.	N.A.	N.A.	N.A.	N.A.	N.A.	N.A.	N.A.	N.A.	N.A.	N.A.	N.A.	N.A.	N.A.	N.A.
Zhou et al., 2021 [28]	187	95	7%	0%	18%	5%	N.A.	N.A.	N.A.	N.A.	N.A.	N.A.	0.5%	0%	3.7%	0%	N.A.	N.A.	N.A.	N.A.

N.A.: not available. EPP: extrapleural pneumonectomy. P/D: pleurectomy/decortication.

## Data Availability

Not applicable.

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
