# Peer review of "Systematic Review and Meta-Analysis of Pleurectomy/Decortication versus Extrapleural Pneumonectomy in the Treatment of Malignant Pleural Mesothelioma"

_jcm, 2022, doi:10.3390/jcm11195544_

Round 1

Reviewer 1 Report

This is a very interesting meta-analysis with a robust methodology. The results are surprising in that pleurectomy combined with decortication has a better survival than extra pleural pneumonectomy. These results are contrary to what is usually accepted. The authors were right to use two methods to estimate 5-year survival. However, to make their conclusions more robust I would suggest to the authors some modifications:

·       Try to provide some explanation of the heterogeneity of the studies if possible, relating to surgical techniques, adjuvant treatments and stages at the time of initiation of treatment .....

·       Perform a sensitivity analysis according to the weight of the studies or adjuvant treatments or methodological quality for example. This is important because we have the impression that a study such as the Flores et al study has an important weight in the outcome of the meta-analysis.

·       The authors should describe the methodological quality of the studies more clearly, as the meta-analysis concerns non-randomized comparative studies, and we would like to know more precisely the possible biases of these studies.

·       There is a publication bias according to figure 3. This point should be taken into account in the discussion.

·       Finally, the tables should be modified to make them more readable.

Author Response

Reviewer 1 highlights some of the critical points of our meta-analysis, we thank the reviewer for the possibility of improving the manuscript.

First comment: Try to provide some explanation of the heterogeneity of the studies if possible, relating to surgical techniques, adjuvant treatments and stages at the time of initiation of treatment .....

Answer: The Reviewer is absolutely right, the high heterogeneity of the studies considered in the meta-analysis emerges not only from the statistical calculations but also from the description of the populations and the procedures made by the various authors. We acknowledge this fact in the limitations section:

“…the choice between P/D and EPP was made on clinical considerations such as patients’ age, comorbidities, pathological characteristics and stage. Each of these parameters had probably an independent impact on survival introducing selection biases. Moreover, it is highly probable that clinical follow-ups were dissimilar in different centers by introducing biases both on survival and on the categorization of complications. We mitigated these discrepancies by statistic methods as best as we can. Furthermore, we should recognize that our meta-analysis includes two different surgical procedures, P/D and extended P/D, each of which may have been performed with different protocols by the different surgical groups. A striking example is the trial published by Lang-Lazdunski and collaborators; the authors state that of the 54 patients subjected to P/D only seven had the resection of the pericardium or diaphragm while five had even typical lung resections.”. “…it is certain that the patients included in this meta-analysis received highly diversified adjuvant and/or neo-adjuvant treatments, since during the studies periods the FDA approved the pemetrexed, first trials of induction therapy appeared around 2010, and immune checkpoint inhibitors were recently introduced in the MPM treatment.”.

We introduced a sentence in order to emphasize also the aspect of the timing as factor of heterogeneity:

Also the lack of indication of the surgical timing as well as the lack of clarification of when the follow-up started certainly contributed to the marked heterogeneity.”

Second comment: Perform a sensitivity analysis according to the weight of the studies or adjuvant treatments or methodological quality for example. This is important because we have the impression that a study such as the Flores et al study has an important weight in the outcome of the meta-analysis.

Answer: We thank the reviewer for raising this issue. Indeed, Flores' study weighs 10.9% in HR random calculation for overall survival. In truth, other studies weigh even more. Indeed, we did a sensitivity test for the main outcome using the "one-leave-out test" (rows 126-127 and 188-189); Reviewer 1 appreciated this method.

Third comment: The authors should describe the methodological quality of the studies more clearly, as the meta-analysis concerns non-randomized comparative studies, and we would like to know more precisely the possible biases of these studies.

Answer: We agree with the Reviewer that the simple classification in Table 1 may not be sufficient. Therefore, we have added the following sentence in the results section:

Ten studies were cohort retrospective studies (15-19, 22-26); of these eight were monocentric studies (16-19, 22-25), one multicentric (15) and one based on a national database (26). Among the ten retrospective studies, four covered a time interval of 20 years or more (15, 16, 22, 23). Two studies were retrospective cohort studies with propensity score matching, both studies were monocentric and covered a time interval between 16 and 19 years (20, 27). A study was a prospective nonrandomized interventional study, performed at a single center. The time interval was only seven years but the decision on which surgery to perform was taken on clinical bases (21). Similarly, clinical criteria were used for the selection of the surgical procedure in all other selected studies.”

Fourth comment: There is a publication bias according to figure 3. This point should be taken into account in the discussion.

Unfortunately, publication bias is extremely frequent in the scientific literature as articles with not significant results are unlikely to be published. Apart from this basic consideration, other well-known factors affect publication bias. We added the following sentence in the discussion:

In addition, a publication bias was noted for the main result (Figure 3), this bias can be attributed to the previously mentioned heterogeneity but also to the quality of the studies, the outcome measures and finally the case.”

Fifth comment: Finally, the tables should be modified to make them more readable.

Please, see the answer to Reviewer 3

Reviewer 2 Report

I really appreciate this manuscript, the message is not original but the work is rigorous and well methodologically performed. The authors should be praised for the clear discussion and for underline also the limitations of their manuscript.

Author Response

We warmly thank the Reviewer 2 for his praise comments to our manuscript.

Reviewer 3 Report

Dear Editor and Authors,

Thank you for asking me to review this work titled “Systematic review and meta-analysis of pleurectomy/decortication versus extrapleural pneumonectomy in the treatment of malignant pleural mesothelioma” by Dr. Danuzzo and her colleagues from the Department of Patho-physiology and Transplantation at the University of Milan in Milan, Italy.

In this review and meta-analysis the authors attempted to review the available literature in order to compare the overall survival and postoperative complications of Extrapleural Pneumonectomy (EPP) and Pleurectomy & Decortication (P/D) and to provide evidence for the best procedure in the treatment of Malignant Pleural Mesothelioma.

This is a very old question in the Thoracic Surgery community and in truth, it has been answered well by a number of studies, not at the MARS 1 randomized study. MARS 2 is currently underway and it should provide even more robust evidence. Therefore, one at first sight may consider this meta-analysis mute and unnecessary. However, after reading the manuscript and evaluating it, it does seem to be well conducted and robust and to offer solid evidence. The authors have followed a clear methodology, have performed a good review of the literature and data mining, have clearly defined end points and a excellent statistical analysis. I was especially intrigued that they used sensitivity analysis/one-leave-out sensitivity test to verify the robustness of their results.Their results are interesting demonstrating a clear advantage of E/D compared to EPP in regards to long term survival (primary outcome),  30-day mortality and perioperative morbidity (secondary outcome).

The manuscript is well written in clear and understandable language and only the figures and tables need some work to make them more clear (see comments below).

Comments:

1.    Table 3 is a bit cluttered and needs some reworking - editing to make it easier to read/understand.

2.     Figure 9 is quite small and difficult to read – please increase the size for the reader.

In conclusion this is a well conducted analysis with a clear message and although there has been quite a lot of work done in the subject matter. I only have some minor edits to offer in regards to tables and figures but apart from that I find the work acceptable. My kindest regards to all and well done.

Author Response

We thank Reviewer 3 for his positive comments. We are sorry for the poor readability of Table 3 and Figure 9. Considering that the originals are very well readable, we believe that the inconvenience is due to the particular format adopted by the Journal for submission to the reviewers. We will ensure that the final version is indeed adequate.

Round 2

Reviewer 1 Report

The authors have responded to the various comments except for the sensitivity analysis. It seems important to me to confirm their results

Otherwise for the rest it is ok for me

Author Response

We regret that we were not sufficiently explanatory in the first round of replies. We performed the sensitivity analysis for the pooled hazard ratio by means one-leave out diagnostics. In particular, we repeatedly fit the meta-analytic model, leaving out one study at a time [29]. The results of this analysis showed the robustness of findings in terms of point estimation, confidence intervals and I2 index, as depicted in attached software screen shoot. This analysis was described with formal sentences in the statistical and result section (rows 138-140)

[29]  Viechtbauer, W. (2021). Model checking in meta-analysis. In C. H. Schmid, T. Stijnen, & I. R. White (Eds.), Handbook of meta-analysis (pp. 219–254). Boca Raton, FL: CRC Press.